# Prediction of Fracture Toughness of Pultruded Composites Based on Supervised Machine Learning

**DOI:** 10.3390/polym14173619

**Published:** 2022-09-01

**Authors:** Radmir Karamov, Iskander Akhatov, Ivan V. Sergeichev

**Affiliations:** Center for Materials Technologies, Skolkovo Institute of Science and Technology, Bolshoy Boulevard 30, bld. 1, 121205 Moscow, Russia

**Keywords:** composite materials, pultrusion, fracture toughness, machine learning, correlation

## Abstract

Prediction of mechanical properties is an essential part of material design. State-of-the-art simulation-based prediction requires data on microstructure and inter-component interactions of material. However, due to high costs and time limitations, such parameters, which are especially required for the simulation of advanced properties, are not always available. This paper proposes a data-driven approach to predicting the labor-consuming fracture toughness based on a series of standard, easy-to-measure mechanical characteristics. Three supervised machine-learning (ML) models (artificial neural networks, a random forest algorithm, and gradient boosting) were designed and tested for the prediction of mechanical properties of pultruded composites. A considerable dataset of mechanical properties was acquired as results of standard tensile, compression, flexure, in-plane shear, and Charpy tests and utilized as the input to predict the fracture toughness. Furthermore, this study investigated the correlations between the obtained mechanical characteristics. Analysis of ML performance showed that fracture toughness had the highest correlations with longitudinal bending and transverse tension and a strong correlation with the longitudinal compression modulus and tensile strength. The gradient boosting decision tree-based algorithms demonstrated the best prediction performance for fracture toughness, with an MSE less than 10% of the average value, providing a prediction within the range of experimental error. The ML algorithms showed potential in terms of determining which macro-level parameters can be used to predict micro-level material characteristics and how. The results provide inspiration for future pultruded composite material design and can enhance the numerical simulations of material.

## 1. Introduction

Prediction of the mechanical properties of novel composite materials is one of the primary goals of research in the field of material design. The prediction of material properties is nowadays attempted by using physics-based simulations: molecular dynamics, finite element methods (FEMs), and others [1,2,3,4]. However, these methods have some limitations due to the unknown microstructure of materials, computational expensiveness, etc. [5,6]. In addition, a propagation error can be introduced in the multiscale modeling of large systems [7,8]. Another way to predict the mechanical properties of materials is the data-driven approach, which has recently become more popular and has given us some inspiration for a novel methodology for the design and characterization of composite materials.

Data-driven approaches based on machine-learning algorithms have been applied in material science in recent decades, accelerating the design and discovery of new functional and structural materials [9]. Machine learning (ML) is a branch of artificial intelligence that allows us to analyze large, noisy datasets and learn and detect data patterns and correlations between input and output variables by optimizing the chosen machine-learning model [10]. This approach is mainly used in the composite materials field to design and discover new materials and their basic properties, such as stiffness and strength.

One of the first works in this field by Mukherjee et al. [11] proposed to predict the yield stress under the tension of metal matrix composites using artificial neural networks with microstructure factors, such as the volume fraction, fiber arrangements, and the properties of the components, as input variables. However, they employed a database that was synthetically generated using the FEM approach to acquire the necessary amount of data due to the insignificant progress in the training algorithms for such models at the time. The authors of [12,13] proposed a methodology to predict composites’ mechanical properties using a very small-sized database. Nevertheless, this work was criticized [14]: the neural networks in their work were too complicated for the given dataset, and highly correlated input and output variables were also used.

In recent years, researchers have worked on predicting the mechanical properties of composite materials based on their inner structure using more advanced machine-learning techniques [15,16,17,18,19,20,21,22]. For instance, the authors of [23] recently applied linear and convolutional neural network (CNN) models to predict the toughness and strength of 2D functional composite systems based on images of their microstructure. They performed calculations using TensorFlow [24], a general-purpose ML framework that has the ability to search for optimal designs with limited information. Yang et al. [25] demonstrated the implementation of a deep-learning, feature-engineering-free approach for predicting the microscale elastic strain field in a given 3D voxel-based microstructure of a high-contrast, two-phase composite. The results showed that deep learning approaches could implicitly learn salient information about local neighborhood details. However, there are still only a limited number of studies that investigate the ML prediction of the advanced mechanical properties of composite materials based on standard, easy-to-measure properties and that discuss their ML correlations without considering the material’s inner structure in detail.

More work on data-driven approaches can be found for other materials but, unfortunately, the published models cannot be applied to predict the mechanical properties of composite materials due to different output parameters and larger datasets. For example, Tiryaki and Aydın [26] applied this methodology to design an artificial neural network model to predict the compression strength of heat-treated wood without comprehensive experiments. The results indicated that the artificial neural network model provided a better prediction than the multiple linear regression model. The strength properties of the heat-treated wood could be determined in a short time with low error rates, allowing the usability of such wood species for structural purposes to be better understood.

One of the most compelling studies [27] was conducted on engineering alloys and described a methodology to acquire elastic, and even plastic, properties based on one sensitive test (instrumented indentation) and the latest developments in deep learning and neural networks. However, this methodology did not consider other machine-learning algorithms, and it is only applicable to instrumented indentation of homogeneous materials and cannot be applied for heterogeneous composite materials. Additionally, the machine-learning correlations between fracture toughness and other material properties were not investigated.

In this study, we propose a new way to predict fracture toughness and to determine its correlations with other mechanical properties based on machine-learning algorithms. We designed and analyzed ML methodologies to predict the mechanical properties of pultruded composites and their correlations throughout the length of the conditionally infinite profile. Pultrusion was selected as it is one of the continuous manufacturing techniques that can provide high-quality and cost-effective composite materials. To characterize a pultruded material with high accuracy, it is necessary to perform a large number of mechanical tests along the produced structural profile, which is a time-consuming and costly task [28,29]. Despite comprehensive studies of pultruded composites performed over the last thirty years (recently reviewed in [30]), there still exists a lack of knowledge about the correlations between different material properties as they are found in the structure. This paper consists of the following parts: Materials and Methods and Results and Discussions. In the following section, the materials and methods used are thoroughly described. In Section 3, the fracture toughness prediction results are presented and the ML-based correlation with easy-to-measure characteristics is discussed.

## 2. Materials and Methods

### 2.1. Pultruded Composite Material 

The material used to illustrate and analyze the machine-learning prediction of the fracture toughness was pultruded glass fiber reinforced polymer (GFRP). The material (Figure 1) was manufactured using a Pultrex P500×6T with a pulling speed of 0.4 m/min and a temperature of 125 °C. 

The material was produced in two days; it was possible to observe deviations in the batches from different days due to different levels of temperature and humidity and human factors.

### 2.2. Mechanical Testing

A 50 m pultruded profile was produced. From each meter, ten specimens for the standard mechanical tests and three specimens for the fracture toughness tests were cut and tested (Figure 2). This allowed us to create a dataset of 50 batches with different properties from along the length of the material that could be used for ML training.

#### 2.2.1. Standard Mechanical Tests

Several standard mechanical properties were chosen to analyze the machine-learning performance with possible correlations with the fracture toughness. The material’s mechanical properties were obtained during a mechanical characterization study consisting of tension, compression, flexure, in-plane shear, and Charpy impact tests. All tests were undertaken in 0° and 90° directions relative to the pulling direction. 

The test standards, testing machine, and obtained properties are given in Table 1.

#### 2.2.2. Fracture Toughness Test

The fracture toughness behavior of pultruded glass fiber reinforced materials has only been described in a few experimental studies [31,32]. In this work, we adopted a wide compact tension method to characterize the transverse fracture properties of the material. This method was proposed by Almeida-Fernandes et al. [33]. The method was proved to be “effective in achieving a stable propagation stage”, and the final properties showed “good agreement across visually based methods”. The test involves applying tensile 20 N/mm loadings to a specially prepared specimen (Figure 3) and observing crack growth. The energy release rate $G_{Ic}^{lam}$ and the stress intensity factor in mode I of loading $K_{Ic}$ (plane strain fracture toughness) were estimated using the following formulae [34]:(1)$$G_{Ic}^{lam}=\frac{K_{Ic}^{2}}{\sqrt{2E_{11}E_{22}}}\;\sqrt{\sqrt{\frac{E_{11}}{E_{22}}}+\frac{E_{11}}{2G_{12}}-\upsilon_{12}},$$
(2)$$K_{Ic}=\frac{P}{t\sqrt{w}}f\left(\frac{a}{w}\right),$$
(3)$$f\left(\frac{a}{w}\right)=\frac{2+\frac{a}{w}}{{\left(1-\frac{a}{w}\right)}^{1.5}}\left[0.886+4.64\left(\frac{a}{w}\right)-13.32{\left(\frac{a}{w}\right)}^{2}+14.72{\left(\frac{a}{w}\right)}^{3}-5.6{\left(\frac{a}{w}\right)}^{4}\right]$$
where $E_{11}$ and $E_{22}$ are the elastic moduli in the crack (longitudinal) and the load (transverse) directions, respectively; $G_{12}$ is the shear elastic modulus; $\upsilon_{12}$ is the Poisson’s ratio; P is the load at a given crack length a; t is the specimen thickness; and w is the geometric parameter of the specimen (see Figure 3).

Fracture toughness experiments take up to ten times longer to complete compared to standard mechanical tests. More detailed information about fracture toughness tests for pultruded GFRP is presented in [33].

We used a stress intensity factor $K_{Ic}$ as a fracture toughness characterization because the energy release rate is highly dependent on the other characteristics.

### 2.3. Machine-Learning Methods

Prediction of one property from others is a regression problem, and the use of machine-learning (ML) algorithms can be presented as a parametric function:(4)y=f(x),
where x and y are vectors for the input and output (to be predicted) data, respectively. ML techniques establish which parameters are used in the function and how they are calculated from the training dataset. The training dataset is represented by input–output vector pairs: $\left(x_{1},y_{1}\right)$, $\left(x_{2},y_{2}\right)$, …, $\left(x_{k},y_{k}\right)$. 

Many ML techniques exist, and, the most popular regression models were implemented in this study; namely, artificial neural networks, a random forest algorithm, and gradient boosting decision trees. We also tried support vector regression and Gaussian process regression but, in this study, they tended to output mean values and were not included in the paper. 

To estimate the ML models’ performances, cross-validation was performed: the models were trained with 90% of the data as training data and 10% as test data. The cross-validation was performed 50 times with different training and test data to acquire statistics for the ML models’ performances.

#### 2.3.1. Artificial Neural Network 

An artificial neural network usually consists of three types of layers: input data, hidden layers, and output data (Figure 4). 

There is always one input and one output layer in a neural network, but the number of hidden layers may vary. Each layer includes neurons that are described by their values. Usually, fully connected neural networks are used for regression problems, where each neuron in the hidden and output layers is connected to all neurons of the previous layer. The values for each neuron are calculated by summing up the values for all neurons from previous layers, multiplied by weights, and adding biases, which can be thought of as analogous to a constant shift in a function. This logic is represented by a set of straight lines in Figure 4. To introduce additional nonlinearity to the algorithm, some nonlinear activation functions (denoted as σ) were applied to each neuron as the overall weight [35]. The mathematical representation of such an algorithm for the calculation of the neuron values $a_{\left(n+1\right)}$ of layer *n +* 1 is the following:(5)$$a_{\left(n+1\right)}=\sigma \left(Wa_{\left(n\right)}+b\right).$$

The parameters we needed to find using the dataset of the input and output values were W weights and b biases. One of the approaches to do that is the backpropagation optimization method, which iteratively updates weights and biases using gradient descent-based algorithms, minimizing the error between the predicted values and the values from the dataset [36].

In this study, a fully connected neural network was used with mean squared error loss. Several NN architectures were tested, the architecture with the best performance consisted of 19 neurons in the input layer; three hidden layers with 20, 40, and 10 neurons; and 1 neuron in the output layer. After selecting seven main features, seven input neurons were used with three hidden layers (7, 14, and 4 neurons), and one neuron in thee output layer. An Adam optimization method was employed [37]. The learning rate was 0.01, and the number of epochs was 3000. The model was implemented in the TensorFlow 2.5 framework.

#### 2.3.2. Random Forest 

The random forest algorithm is an advanced ensemble algorithm [38]. The fundamental elements of this algorithm are decision trees (Figure 5). For regression problems, a decision tree can be represented as a function created by recursively partitioning each independent variable. Based on the partition, the output value is predicted. The partitioning is performed to optimize the mean squared errors (MSEs) between the predicted and actual values. The optimal tree is the smallest tree that has the minimum relative error with cross-validation.

The random forest regression algorithm consists of N (usually more than 100) independent decision trees with a random subset of independent variables. To render the decision trees in the random forest algorithm independent, the bagging method [39] was used. A machine-learning model with robust resistance to overfitting was created by combining a random subset of input variables and random training-set selection. The final prediction of a random forest model is an unweighted average of the predictions of all the individual trees.

Although random forest models cannot be graphically presented as decision trees, the variable importance measures (VIMs) can be calculated [40,41]. In the VIM concept, the impact of optimizing an input variable partition is measured with respect to the change in the MSE. The greater the prediction accuracy reduction is during optimization, the more significant the impact of this variable is on the random forest model. Variable importance measures can be considered as machine-learning correlations between input and output parameters.

In this work, we implemented a random forest algorithm from the scikit-learn library [42] for Python. The number of trees was 100; other parameters were left as default, which gave the best performance.

#### 2.3.3. Gradient Boosting

Gradient boosting decision tree methods [43] are also advanced ensemble algorithms that can be based on decision trees (Figure 5). This technique is based on robust agglomeration of additive weak learning models, which iteratively complement each other. The training process of additive models can be represented as follows:(6)$$F_{m}\left(x\right)=F_{m-1}\left(x\right)+\alpha h_{m}\left(x\right)$$

Here, $F_{m}\left(x\right)$ is an agglomeration of models, composed of m weak learning models $h_{m}\left(x\right)$ (i.e., basis functions, in our case, decision trees), that corrects the error of the $F_{m-1}\left(x\right)$. To mitigate the model’s overfitting, a scaling factor α is applied, which can vary from 0 to 1 in order to decrease the contribution of each iteration. In gradient tree boosting regression, the basic functions $h_{m}\left(x\right)$ are represented by small regression trees. Recently, the gradient tree boosting method has been regularized and implemented in the extreme gradient tree boosting algorithm (XGBoost) [44]. The most significant improvements implemented in XGBoost include loss function regularization and normalization, Taylor expansion enhancement of the loss function, and a more complex split-finding algorithm for many features. 

In this study, we used XGBoost Python libraries with 100 iterative, weak learning algorithms; a scaling factor of 0.1; and a learning rate of 0.2. The other parameters were left as default.

#### 2.3.4. Evaluation Criteria

To evaluate the model performance, the root mean square error (*RMSE*) and the mean absolute error (*MAE*) were used:(7)$$RMSE=\sqrt{\frac{1}{N}\overset{N}{\underset{i=1}{\sum }}{\left(y_{i}-{\hat{y}}_{i}\right)}^{2}}$$
(8)$$MAE=\frac{1}{N}\overset{N}{\underset{i=1}{\sum }}\left|y_{i}-{\hat{y}}_{i}\right|$$
where *N* is the number of performed tests, and $y_{i}$ and $\hat{y_{i}}$ are real and predicted values, respectively.

In addition, the coefficient of determination (also called $R^{2}$) was used to evaluate the model performance:(9)$$R^{2}=1-\frac{\sum_{i=1}^{n}{\left(y_{i}-{\hat{y}}_{i}\right)}^{2}}{\sum_{i=1}^{n}{\left(y_{i}-\bar{y}\right)}^{2}}$$
where $\bar{y}$ is the mean value of the true data. The coefficient of determination shows how much of the variability in true values can be caused by the relationship to the predicted values. $R^{2}$ is represented by a value up to 1: $R^{2}=1$ means the model predicts data with a perfect fit, with 0 the model always predicts the mean value, and negative values mean the model cannot predict the data.

The Pearson method was used for correlation calculations. Pearson’s correlation coefficient is defined as:(10)$$r_{xy}=\frac{\sum_{i=1}^{n}\left(x_{i}-\bar{x}\right)\left(y_{i}-\bar{y}\right)}{\sum_{i=1}^{n}{\left(x_{i}-\bar{x}\right)}^{2}\sum_{i=1}^{n}{\left(y_{i}-\bar{y}\right)}^{2}}$$
where n is sample size, $x_{i}\;$and $y_{i}$ are the individual sample points, and $\bar{x}$ and $\bar{y}$ are the sample means. Pearson’s correlation coefficient ranges from −1 to 1, where −1 and 1 indicate perfectly negative and positive linear correlations, respectively, and 0 indicates no linear correlation. 

To compare Pearson’s correlation coefficients with the variable importance measures (which range from 0 to 1; the sum of VIMs gives 1), the fracture toughness correlation coefficients were normalized by dividing the coefficients by their sum and taking the absolute value:(11)$$r_{xy_{i}}^{norm}=\frac{\left|r_{xy_{i}}\right|}{\sum_{i=1}^{n}r_{xy_{i}}}$$

## 3. Results and Discussions

Overall, to investigate ML prediction capabilities and property correlations, 50 batches of specimens were obtained, 600 specimens were tested, and 900 properties were extracted. The statistics of the mechanical test results are presented in Table 2. The coordinates of the specimen location were also treated as an input parameter, and they were considered as a variation in the environment parameters and as resulting from the human factor during the manufacturing process. Even with this extensive work completed, the dataset would be considered small in the machine-learning field.

The properties were chosen according to their low acquisition time, low costs, and possible correlations with fracture toughness. Fractures in composite materials are usually associated with the matrix properties [45]. We expected a high correlation with bending, impact, and tensile (only in the transverse direction) properties, both for the modulus and strength, because these characteristics are also associated with matrix properties. Other properties were considered for verification of the previous research and hypotheses.

The analysis of mechanical properties indicated anisotropic behavior, which is typical for pultruded materials: the performance at 0° (fiber or longitudinal direction) was much high than at 90° (transverse direction). The material was characterized by relatively high strength (properties 9 and 15) and low Young’s modulus, especially in the transverse direction (properties 4, 12, and 18). There were property deviations along the length (due to manufacturing variances), which allowed us to train the ML models on a variable dataset. 

Standard deviations of properties were calculated for the obtained dataset. These deviations were mostly observed along the length of the profile, while local deviations were minimal. This was most likely caused by deviations in the conditions for composite production and the human factor. The deviation in fracture toughness was relatively small, considering the measurement difficulties described in Section 2.2.2, and the deviations were lower than those for other fracture toughness measurements due to the experiment’s specifics, as explained in [32]. These deviations were also mostly introduced due to the property differences along the length of the material, and low deviations were observed in one batch (the batch of one meter length). In addition, deviations were considered beneficial for our purposes, since it gave the machine learning more diverse data to learn from.

For additional analysis, the correlation coefficients, representing the relationship between two variables, were obtained using Pearson’s method. The heatmap of correlation coefficients for the mechanical properties is shown in Figure 6. 

As expected, the fracture toughness stress intensity factor had especially strong correlations with the elastic bending (property 2) and tensile elasticity (property 18) properties, which are associated with matrix properties. Interestingly, the stress intensity factor also had noticeable correlations with the shear strength (properties 7 and 8), the compression modulus in the longitudinal direction (property 10), the compression strength in the transverse direction (property 11), and the tensile strength and modulus in the longitudinal direction (properties 15 and 16). These correlations describe the performance of the material overall and will be discussed with ML variable importance measures. Other properties had low or no correlations with the stress intensity factor.

Three selected machine-learning algorithms were employed: neural networks, a random forest algorithm and XGBoost. Other machine-learning algorithms, such as support vector regression and Gaussian processes (kriging), were not originally designed to handle multidimensional data, and they performed worse in our case. The selected machine-learning models, described in the Materials and Methods section, were trained on the dataset. We used a sensitivity analysis for optimal training procedure where different learning hyperparameters were found for the best prediction results. To test the machine learning methods, we used cross-validation for 45 specimen results for training and 5 specimen results for validation (Figure 7d). The cross-validation was repeated with the entire dataset, and the combined results are presented in Figure 7 and Table 3. 

As shown in the plots (Figure 7), ensemble-based machine learning methods (random forest, XGBoost) predicted fracture toughness with high accuracy, with 9.8% and 9.4% errors in relation to the mean experimental results, which were within the experimental error of 12.7%. The accuracy was also confirmed by an $R^{2}$ coefficient above 0.5. Despite thorough selection of the architecture, neural networks showed the worst performance due to the small dataset used; it was challenging to optimize all the NN parameters when we did not have enough data. Random forest and XGBoost showed similar promising results. XGBoost slightly outperformed random forest in the RMSE, MAE, and $R^{2}$ metrics.

Random forest and XGBoost allow the calculation of variable importance measures (Figure 8), which show how the input feature influence the results during the optimization process.

Variable importance measure analysis showed that random forest and XGBoost used different variables in their predictions: random forest mostly leaned on the bending modulus in the 0° direction (property 2), while XGBoost leaned on the tensile modulus in the 90° direction (property 18). 

It should be noted that both algorithms, random forest and XGBoost, considered the longitudinal compression modulus and tensile strength (properties 10 and 15) as the second most important variables to predict fracture toughness, despite their having the same correlations as many other characteristics (for example, properties 7 and 11). The influence of the compression modulus and tensile strength on fracture properties is not straightforward or intuitive. However, the influence of compression and tension can be explained by local strain/stress fields around fibers during crack development. Our theory is that, during the test, before energy is released and the crack grows, energy accumulates at locations under compression. This hypothesis can be confirmed by consulting previous studies; for instance, those by Tsouvalis et al. [46] or Song et al. [47], where the authors discuss the fiber–matrix interface and observe compression strain and stress fields during fracture simulation. This led us to the assumption that machine learning can find mechanical correlations at the micro-level, despite only knowing the macro-characteristics. However this theory needs to be researched further and the machine-learning transition from macro- to micro-parameters needs to be specifically investigated.

To further investigate the ML prediction performance and the correlations of the fracture toughness and standard properties, we selected the five most significant properties from the analysis of both variable importance measures; for the random forest: 0, 2, 5, 10, and 15; for XGBoost: 2, 7, 10, 15, and 18. Considering that some properties were repeated, overall, we selected 0, 2, 5, 7, 10, 15, and 18, which had high or moderate correlations with the fracture toughness stress intensity factor.

With these seven properties, the machine-learning algorithms were trained again, and the results are presented in Figure 9 and Table 4.

With the selected properties and optimized architecture, the neural network’s performance strongly increased. Therefore, we could exploit a simpler architecture for fewer features, which would be trained better on small datasets. However, the prediction accuracies of random forest and XGBoost fell slightly, especially the accuracy of the results for XGBoost because, according to its VIMs, the algorithm used almost all the properties for precise predictions and dropped properties could not be ignored. It is worth noting that, with the selected features, all the ML algorithms predicted the fracture toughness within the experimental error range.

In the future, the presented approach will be extended for predictions and correlation analysis of different mechanical properties and different composite materials. A larger dataset will be acquired from different manufacturers, other production methods, and composite components. The correlations revealed here will be verified, and a pre-trained model for the general case of composite materials will be developed. Furthermore, the present methodology will be applied to cut back the mechanical testing study required for the comprehensive characterization of composite materials.

## 4. Conclusions

This paper proposes a data-driven approach for prediction of the fracture toughness mechanical property based on other properties of the material. Furthermore, correlations of easy-to-measure mechanical properties with fracture toughness were investigated based on machine-learning algorithms. To illustrate the proposed approach, three machine-learning models were implemented and trained: an artificial neural network, a random forest algorithm, and gradient boosting decision trees. A considerable dataset (900 properties, 50 batches) of mechanical properties for a pultruded composite material was obtained to train the models. 

Machine learning proved its ability to predict fracture toughness behavior in the absence of information about the inner microstructure. The analysis of the ML models showed that the gradient boosting model predicted the stress intensity factor with an MSE less than 10%, which was equivalent to the experimental error. The random forest algorithm showed similar performance. Prediction of fracture toughness with the neural network method was considered statistically unsatisfactory due to the small database used for the significant number of trainable parameters that had to be optimized during training. 

Feature selection and correlation analysis showed that some properties correlated with the material’s fracture toughness more than others: elastic bending and tensile elasticity correlated well with fracture toughness, as expected, due to the nature of the matrix behavior, and a good correlation with fracture toughness was also observed for the longitudinal compression modulus and tensile strength, which could have been caused by energy accumulations at locations under compression. However, future investigation is required. Machine learning only with selected features with high correlations significantly improved the neural network predictions but lowered the accuracy of the ensemble-based algorithms. 

Overall, machine-learning algorithms show potential for determining which mechanical characteristics at the micro-level are correlated with the macro-parameters without knowing the internal microstructure. The data-driven approach for mechanical property prediction can complement and enhance physics-based prediction methods and lead to a cut back in the experimental framework required to characterize composite material extensively for structural applications.

## Figures and Tables

**Figure 1 polymers-14-03619-f001:**
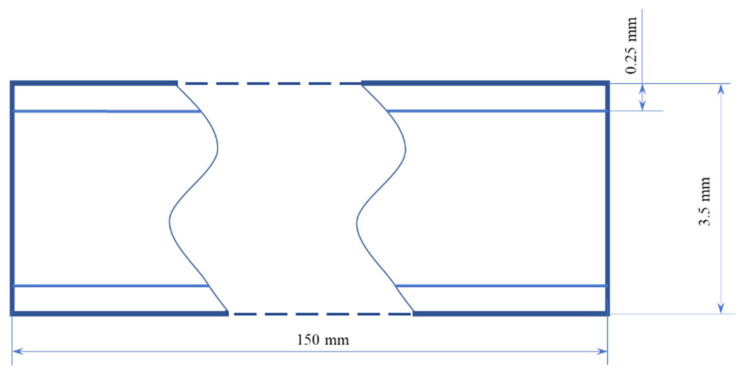
Pultruded profile side cross-section (side view).

**Figure 2 polymers-14-03619-f002:**
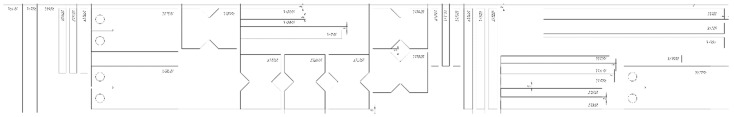
Cutting scheme for specimens from one meter of the pultruded profile (the specimen numbers are indicated for internal use).

**Figure 3 polymers-14-03619-f003:**
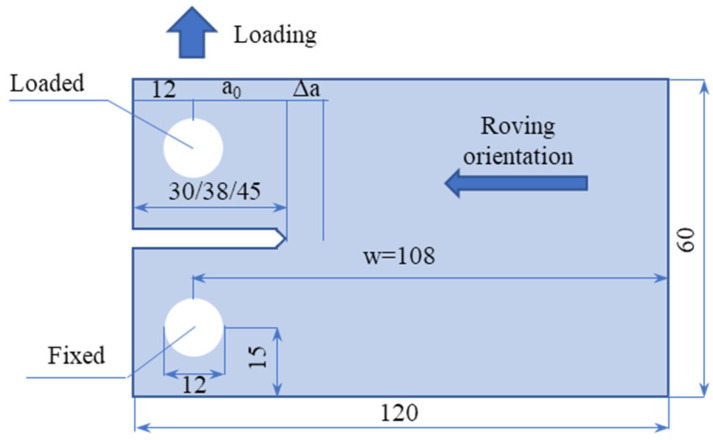
Specimen geometry for the fracture toughness test.

**Figure 4 polymers-14-03619-f004:**
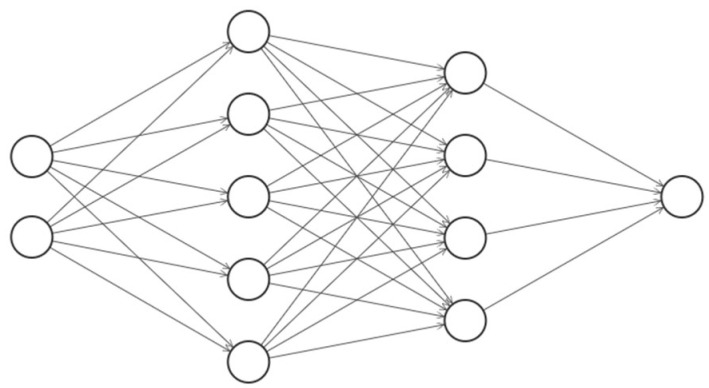
Example of fully connected neural network with two input neurons (input layer), nine hidden neurons (two hidden layers), and one output neuron (output layer).

**Figure 5 polymers-14-03619-f005:**
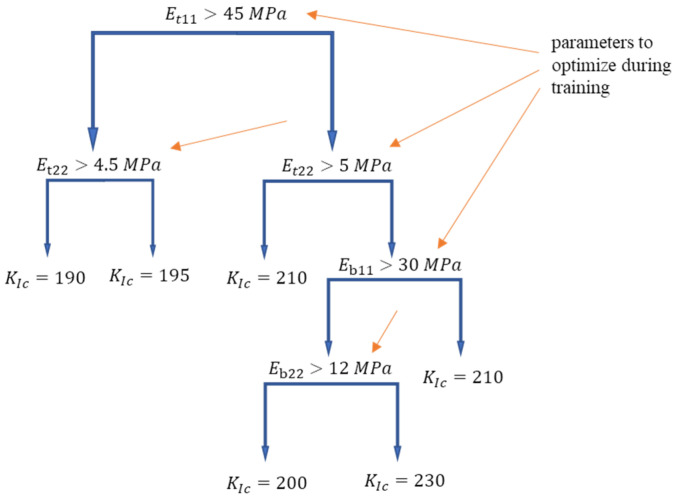
Example of one regression decision tree with depth equal to 4.

**Figure 6 polymers-14-03619-f006:**
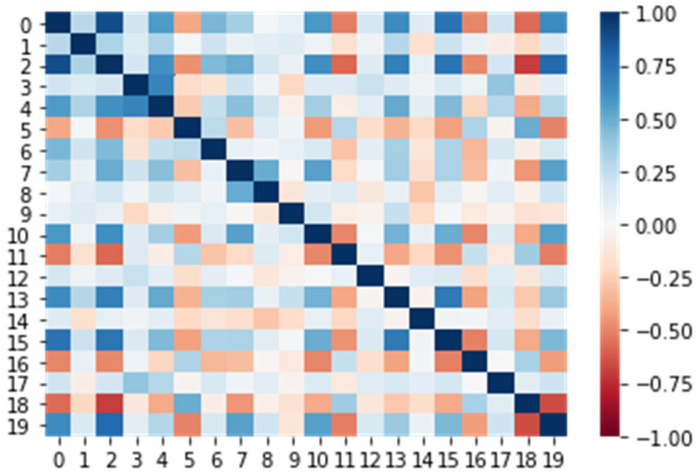
Heatmap of correlations in the acquired dataset. The last row (or column) represents the correlation of the fracture toughness with other properties. Property numeration is the same as in Table 2.

**Figure 7 polymers-14-03619-f007:**
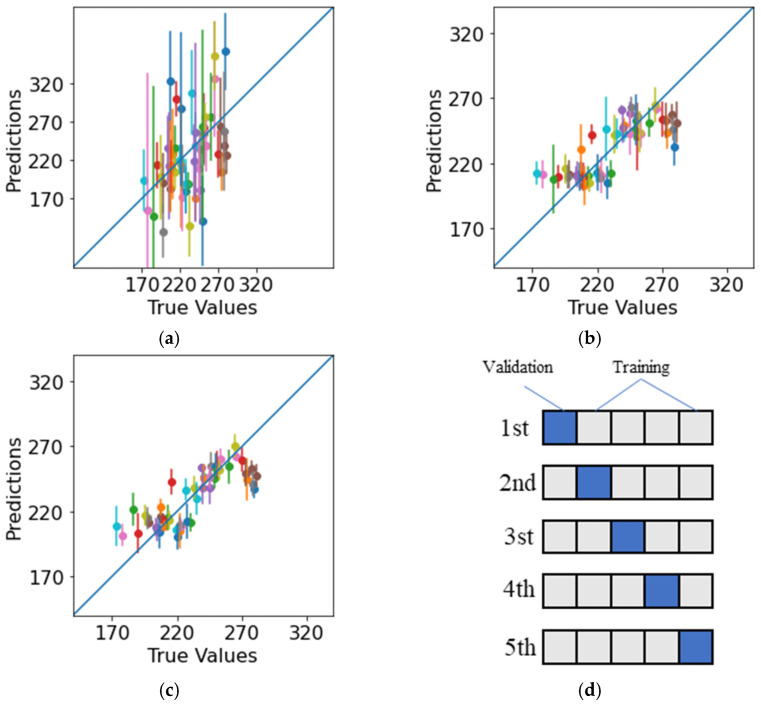
Plots for predicted vs. true values with dots indicating the mean prediction for one data point and vertical lines indicating error bars across different training sessions during cross-validation: (**a**) neural network (note the different scale for the presentation of error bars), (**b**) random forest, (**c**) XGBoost, (**d**) cross-validation scheme.

**Figure 8 polymers-14-03619-f008:**
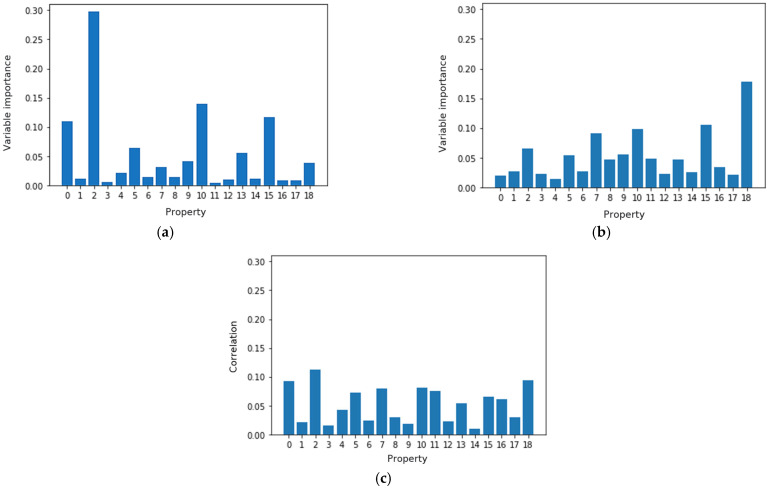
Variable importance vs. normalized correlations: (**a**) random forest, (**b**) XGBoost, (**c**) normalized correlations. Properties are assigned similarly as in Table 2.

**Figure 9 polymers-14-03619-f009:**
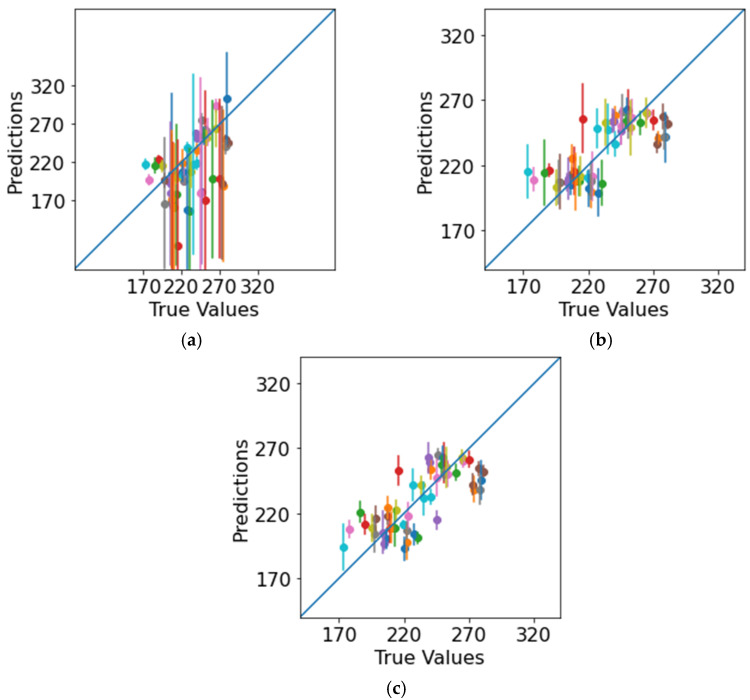
Predicted vs. true plots for selected features, with dots indicating the mean predictions for individual data points and vertical lines indicating error bars across different training sessions during cross-validation: (**a**) Neural network (note the different scale for the presentation of the error bars), (**b**) random forest, (**c**) XGBoost.

**Table 1 polymers-14-03619-t001:** Summary of mechanical tests.

Mechanical Test	Standard	Machine	Strain Measurements	Output Properties
Tension	ASTM D3039	Instron 5969	DIC	Tensile modulus, strength
Compression	ASTM D6641 CLC			Compression modulus, strength
Flexure	ASTM D7264			Flexure modulus, strength
In-plane shear	ASTM D7078			Shear strength, shear modulus
Charpy impact	ASTM D256	Instron CEAST 9340	Strain gauge 64k	Failure energy

**Table 2 polymers-14-03619-t002:** Statistics and numbering of the analyzed mechanical properties.

#	Property	Min	Max	Mean	STD
0	Coordinates, m	0	50	-	-
1	0° bending failure stress, MPa	239	676	559.2	71.3
2	0° bending modulus, GPa	25.3	34.6	30.3	3.2
3	90° bending failure stress, MPa	14.7	172.0	132.5	25.9
4	90° bending modulus, GPa	5.1	13.8	11.8	1.3
5	0° shear strength, MPa	24.2	58.1	45.6	7.9
6	0° shear modulus, GPa	4.3	9.3	6.9	1.2
7	90° shear strength, MPa	45.2	69.2	54.4	4.9
8	90° shear modulus, GPa	4.9	9.9	7.8	0.7
9	0° compression strength, MPa	366	660	494.4	58.0
10	0° compression modulus, GPa	34.3	54.2	47.4	3.9
11	90° compression strength, MPa	71.6	117.0	91.9	10.1
12	90° compression modulus, GPa	5.2	9.1	6.7	0.8
13	0° impact failure energy, kJ/m^2^	30	142	70	27
14	90° impact failure energy, kJ/m^2^	9.5	22.5	14.5	3
15	0° tensile strength, MPa	434	839	657.5	85.6
16	0° tensile modulus, GPa	41.9	49.3	45.1	1.5
17	90° tensile strength, MPa	36.4	57.4	45.6	4.5
18	90° tensile modulus, GPa	4.2	6.6	5.4	0.5
19	0° KIC for 40 mm	173.0	305.8	232.4	29.6

**Table 3 polymers-14-03619-t003:** Prediction results for fracture toughness; the error in relation to the mean experimental results is in brackets.

Model	Mean Prediction RMSE	Mean Prediction MAE	$\mathrm{Mean}\;R^{2}$
Neural network	49.1 (21%)	35.0 (15%)	−1.92
Random forest	22.7 (9.8%)	17.0 (7.3%)	0.58
XGBoost	21.8 (9.4%)	16.0 (6.9%)	0.61

**Table 4 polymers-14-03619-t004:** Prediction results for selected features of fracture toughness; the error in relation to the mean experimental results is in brackets.

Model	Mean Prediction RMSE	Mean Prediction MAE	$\mathrm{Mean}\;R^{2}$
Neural network	28.4 (12.2%)	24.0 (10.3%)	−1.15
Random forest	23.4 (10.1%)	17.5 (7.5%)	0.52
XGBoost	23.8 (10.2%)	18.8 (8.1%)	0.54

## Data Availability

The data used in this research will be made available on request.

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
