# Peer review of "Prediction of Fracture Toughness of Pultruded Composites Based on Supervised Machine Learning"

_polymers, 2022, doi:10.3390/polym14173619_

Round 1
Reviewer 1 Report
This manuscript uses machine learning algorithms to obtain fracture toughness of pultruded composites. Authors could show good prediction of fracture toughness based on three different algorithms: Neural network, Random forest and Gradient boosting. The authors show a very interesting work which has benefits for reducing the experimental tests to characterise mechanical properties and the expensive costs, but the manuscript lacks some information which minor revisions are needed as follows:
1. The introduction should also point out to the novelty of the current study, and it should consist of the structure of the paper.
2. On page 4, specify these parameters (p, w, t and a) in Figure 3. For example, parameter w is not clear in this figure.
3. On page 5, section 2.3.1, It seems authors are referring to Figure 3 but, in the manuscript, it is written “Figure 1”. Please check it.
4. On page 10, Figure 8.c, the normalized correlations have not been described inside the manuscript. It is suggested to add some explanations for it.
5. On page 12, in section 5, authors mentioned that the MSE error for gradient boosting method is less than 10% and random forest showed similar performance, but this information has not been provided in the results. Please add them in the results section.
6. How do authors estimate the performance of other machine learning algorithms such as support vector regression and gaussian process in this application?
Author Response
Thank you for your comments and feedback!
Please see the attachment for the responses.

Reviewer 2 Report
Please see the attachment.

Author Response
Thank you very much for your constructive feedback and thoughtful comments!
Please see the attachment for the responses.

Reviewer 3 Report
Good work.
Author Response
Thank you for your feedback!
Round 2
Reviewer 2 Report
The authors have adequately addressed the points raised in my review. The manuscript has been improved and I believe it is ready for publication.